# CLIMATE-SENSITIVE URBAN PLANNING THROUGH OPTIMIZATION OF TREE PLACEMENTS

## ABSTRACT

Climate change is increasing the intensity and frequency of many extreme weather events, including heatwaves, which results in increased thermal discomfort and mortality rates. While global mitigation action is undoubtedly necessary, so is climate adaptation, e.g., through climate-sensitive urban planning. Among the most promising strategies is harnessing the benefits of urban trees in shading and cooling pedestrian-level environments. Our work investigates the challenge of optimal placement of such trees. Physical simulations can estimate the radiative and thermal impact of trees on human thermal comfort but induce high computational costs. This rules out optimization of tree placements over large areas and considering effects over longer time scales. Hence, we employ neural networks to simulate the point-wise mean radiant temperatures–a driving factor of outdoor human thermal comfort–across various time scales, spanning from daily variations to extended time scales of heatwave events and even decades. To optimize tree placements, we harness the innate local effect of trees within the iterated local search framework with tailored adaptations. We show the efficacy of our approach across a wide spectrum of study areas and time scales. We believe that our approach is a step towards empowering decision-makers, urban designers and planners to proactively and effectively assess the potential of urban trees to mitigate heat stress.

## 1 INTRODUCTION

Climate change will have profound implications on many aspects of our lives, ranging from the quality of outdoor environments and biodiversity, to the safety and well-being of the human populace (United Nations, 2023). Particularly noteworthy is the observation that densely populated urban regions, typically characterized by high levels of built and sealed surfaces, face an elevated exposure and vulnerability to heat stress, which in turn raises the risk of mortality during heatwaves (Gabriel & Endlicher, 2011). The mean radiant temperature ($T_{mrt}$, °C) is one of the main factors affecting daytime outdoor human thermal comfort (Holst & Mayer, 2011; Kántor & Unger, 2011; Cohen et al., 2012).[1] High $T_{mrt}$ can negatively affect human health (Mayer et al., 2008) and $T_{mrt}$ has a higher correlation with mortality than air temperature (Thorsson et al., 2014). Consequently, climate-sensitive urban planning should try to lower maximum $T_{mrt}$ as a suitable climate adaption strategy to enhance (or at least maintain) current levels of outdoor human thermal comfort.

Among the array of climate adaption strategies considered for mitigation of adverse urban thermal conditions, urban greening, specifically urban trees, have garnered significant attention due to their numerous benefits, including a reduction of $T_{mrt}$, transpirative cooling, air quality (Nowak et al., 2006), and aesthetic appeal (Lindemann-Matthies & Brieger, 2016). Empirical findings from physical models have affirmed the efficacy of urban tree canopies in improving pedestrian-level outdoor human thermal comfort in cities (De Abreu-Harbich et al., 2015; Lee et al., 2016; Chàfer et al., 2022). In particular, previous studies found the strong influence of tree positions (Zhao et al., 2018; Abdi et al., 2020; Lee et al., 2020). Correspondingly, other work has studied the optimization of tree placements, deploying a wide spectrum of algorithms, such as evolutionary, greedy, or hill climbing algorithms (Chen et al., 2008; Ooka et al., 2008; Zhao et al., 2017; Stojakovic et al., 2020; Wallenberg et al., 2022). However, these works were limited by the computational cost of physical models, which rendered the optimization of tree placements over large areas or long time scales infeasible.

---

[1] $T_{mrt}$ is introduced in more detail in Appendix A.

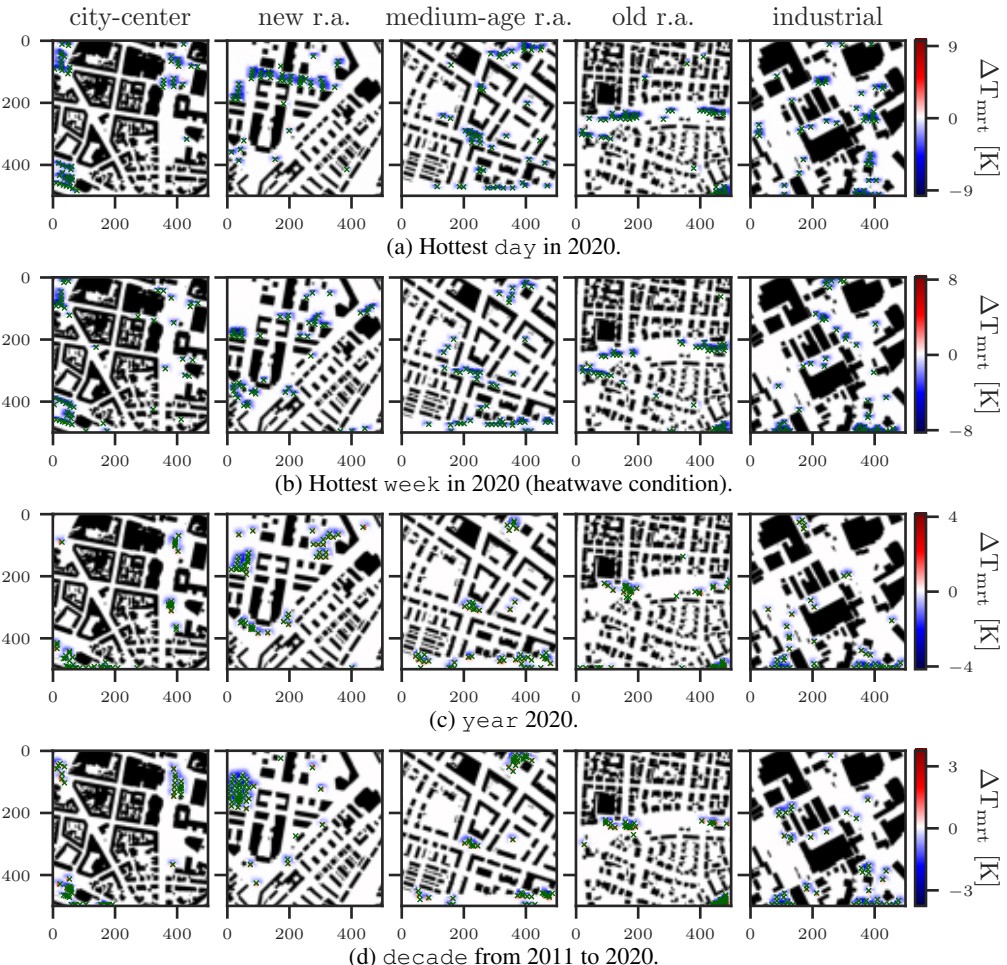

Figure 1: Optimizing tree placements can substantially reduce point-wise $T_{mrt}$, e.g., during heatwaves, leading to improved outdoor human thermal comfort. Optimized placements of 50 added trees (green crosses), each with a height of $12\,m$ and crown diameter of $9\,m$, for the hottest day (1(a)) and week in 2020 (1(b), the entire year 2020 (1(c)), and the entire decade from 2011 to 2020 (1(d)) across diverse urban neighborhoods (from left to right: city-center, recently developed new r.a. (residential area), medium-age r.a., old r.a., industrial area).

Recently, there has been increased interest in applications of machine learning in climate science (Rolnick et al., 2022). For example, Briegel et al. (2023) and Huang & Hoefler (2023) improved the computational efficiency of modeling and data access, respectively. Other works sought to raise awareness (Schmidt et al., 2022), studied the perceptual response to urban appearance (Dubey et al., 2016), or harnessed machine learning as a means to augment analytical capabilities in climate science (e.g., Albert et al. (2017); Blanchard et al. (2022); Teng et al. (2023); Otness et al. (2023)). Besides these, several works used generative image models or reinforcement learning for urban planning, e.g., land-use layout (Shen et al., 2020; Wang et al., 2020; 2021; 2023; Zheng et al., 2023). Our work deviates from these prior works, as it directly optimizes a meteorological quantity ($T_{mrt}$) that correlates well with heat stress experienced by humans (outdoor human thermal comfort).

In this work, we present a simple, scalable yet effective optimization approach for positioning trees in urban environments to facilitate *proactive climate-sensitive planning* to adapt to climate change in cities.[2] We harness the iterated local search framework (Lourenço et al., 2003; 2019) with tailored adaptations. This allows us to efficiently explore the solution space by leveraging the inherently local influence of individual trees to iteratively refine their placements. We initialize the search with a simple greedy heuristic. Subsequently, we alternately perturb the current best tree placements with a genetic algorithm (Srinivas & Patnaik, 1994) and refine them with a hill climbing algorithm.

---

[2]Code is available at https://anonymous.4open.science/r/tree-planting.

To facilitate fast optimization, we use a U-Net (Ronneberger et al., 2015) as a computational shortcut to model point-wise $T_{\mathrm{mrt}}$ from spatio-temporal input data, inspired by Briegel et al. (2023). However, the computational burden for computing aggregated, point-wise $T_{\mathrm{mrt}}^{M,\phi}$ with aggregation function $\phi$, e.g., mean, over long time periods $M$ with $|M|$ meteorological (temporal) inputs is formidable, since we would need to predict point-wise $T_{\mathrm{mrt}}$ for all meteorological inputs and then aggregate them. To overcome this, we propose to instead learn a U-Net model that directly estimates the aggregated, point-wise $T_{\mathrm{mrt}}^{M,\phi}$, effectively reducing computational complexity by a factor of $\mathcal{O}(|M|)$. Lastly, we account for changes in the vegetation caused by the positioning of the trees, represented in the digital surface model for vegetation, by updating depending spatial inputs, such as the sky view factor maps for vegetation. Since conventional protocols are computationally intensive, we learn an U-Net to estimate the sky view factor maps from the digital surface model for vegetation.

Our evaluation shows the efficacy of our optimization of tree placements as a means to improve outdoor human thermal comfort by decreasing point-wise $T_{\mathrm{mrt}}$ over various time periods and study areas, e.g., see Figure 1. The direct estimation of aggregated, point-wise $T_{\mathrm{mrt}}^{M,\phi}$ yields substantial speed-ups by up to 400,000x. This allows for optimization over extended time scales, including factors such as seasonal dynamics, within large neighborhoods ($500\,\mathrm{m}$ x $500\,\mathrm{m}$ at a spatial resolution of $1\,\mathrm{m}$). Further, we find that trees' efficacy is affected by both daily and seasonal variation, suggesting a dual influence. In an experiment optimizing the placements of existing trees, we found that alternative tree placements would have reduced the total number of hours with $T_{\mathrm{mrt}} > 60\,^\circ\mathrm{C}$–a recognized threshold for heat stress (Lee et al., 2013; Thorsson et al., 2017)–during a heatwave event by a substantial $19.7\,\%$. Collectively, our results highlight the potential of our method for climate-sensitive urban planning to *empower decision-makers in effectively adapting cities to climate change*.

## 2  DATA

Our study focuses on the city of CITY. [3] Following Briegel et al. (2023), we used spatial (geometric) and temporal (meteorological) inputs to model point-wise $T_{\mathrm{mrt}}$. The spatial inputs include: digital elevation model; digital surface models with heights of ground and buildings, as well as vegetation; land cover class map; wall aspect and height; and sky view factor maps for buildings and vegetation. Spatial inputs are of a size of $500\,\mathrm{m}$ x $500\,\mathrm{m}$ with a resolution of $1\,\mathrm{m}$. Raw LIDAR and building outline (derived from CityGML with detail level of 1) data were provided by the City of CITY (2018; 2021) and pre-processed spatial data were provided by Briegel et al. (2023). We used air temperature, wind speed, wind direction, incoming shortwave radiation, precipitation, relative humidity, barometric pressure, solar elevation angle, and solar azimuth angle as temporally varying meteorological inputs. We used past hourly measurements for training and hourly ERA5 reanalysis data (Hersbach et al., 2020) for optimization. Appendix B provides more details and examples.

## 3  METHODS

We consider a function $f_{T_{\mathrm{mrt}}}(s,\ m)$ to model point-wise $T_{\mathrm{mrt}} \in \mathbb{R}^{h \times w}$ of a spatial resolution of $h \times w$. It can be either a physical or machine learning model and operates on a composite input space of spatial $s = [s_v,\ s_{\neg v}] \in \mathbb{R}^{|S| \times h \times w}$ and meteorological inputs $m \in M$ from time period $M$, e.g., heatwave event. The spatial inputs $S$ consist of vegetation-related $s_v$ (digital surface model for vegetation, sky view factor maps induced by vegetation) and non-vegetation-related spatial inputs $s_{\neg v}$ (digital surface model for buildings, digital elevation model, land cover class map, wall aspect and height, sky view factor maps induced by buildings). Vegetation-related spatial inputs $s_v$ are further induced by the positions $T_p \in \mathbb{N}^{k \times h \times w}$ and geometry $t_g$ of $k$ trees by function $f_v(t_p,\ t_g)$. During optimization we simply modify the digital surface model for vegetation and update depending spatial inputs accordingly (see Section 3.3). To enhance outdoor human thermal comfort, we want to minimize the aggregated, point-wise $T_{\mathrm{mrt}}^{M,\phi} \in \mathbb{R}^{h \times w}$ for a given aggregation function $\phi$, e.g., mean, and time period $M$ by seeking the tree positions

$$t_p^* \in \arg\min_{t_p'} \phi(\{f_{T_{\mathrm{mrt}}}([f_v(t_p',\ t_g),\ s_{\neg v}],\ m) \mid \forall m \in M\}) \qquad , \qquad (1)$$

in the urban landscape, where we keep tree geometry $t_g$ fixed for the sake of simplicity.

---

[3]The placeholder ensures double blindness and will be replaced upon acceptance.

---

**Algorithm 1** Iterated local search to find the best tree positions.

1: **Input:** $\Delta T_{\mathrm{mrt}}^t$, $f_{T_{\mathrm{mrt}}^{M,\phi}}$, number of trees $k$, number of iterations $I$, local optima buffer size $b$
2: **Output:** best found tree $s_*$ in $S_*$
3: $s_* \leftarrow \texttt{TopK}(\Delta T_{\mathrm{mrt}}^t,\ k)$          # Initialization
4: **for** $i = 1, \ldots, I$ **do**
5:     $s' \leftarrow \texttt{PerturbationWithGA}(S_*, \Delta T_{\mathrm{mrt}}^t)$          # Perturbation
6:     $s_*' \leftarrow \texttt{HillClimbing}(s')$          # Local search
7:     $S_* \leftarrow \texttt{TopK}(\{f_{T_{\mathrm{mrt}}^{M,\phi}}(s)|s \in S_* \cup s_*'\},\ b)$        # Acceptance criterion
8: **end for**

---

Numerous prior works (Chen et al., 2008; Ooka et al., 2008; Zhao et al., 2017; Stojakovic et al., 2020; Wallenberg et al., 2022) have tackled above optimization problem. Nevertheless, these studies were encumbered by formidable computational burdens caused by the computation of $T_{\mathrm{mrt}}$ with conventional (slow) physical models, rendering them impractical for applications to more expansive urban areas or extended time scales, e.g., multiple days of a heatwave event. In this work, we present both an effective optimization method based on the iterated local search framework (Lourenço et al., 2003; 2019) (Section 3.1, see Algorithm 1 for pseudocode), and a fast and scalable approach for modeling $T_{\mathrm{mrt}}$ over long time periods (Sections 3.2 and 3.3, see Figure 2 for an illustration).

## 3.1 OPTIMIZATION OF TREE PLACEMENTS

To search tree placements, we adopted the iterated local search framework from Lourenço et al. (2003; 2019) with tailored adaptations to leverage that the effectiveness of trees is bound to a local neighborhood. The core principle of iterated local search is the iterative refinement of the current local optimum through the alternation of perturbation and local search procedures. We initialize the first local optimum by a simple greedy heuristic. Specifically, we compute the difference in $T_{\mathrm{mrt}}$ ($\Delta T_{\mathrm{mrt}}^t$) resulting from the presence or absence of a single tree at every possible position on the spatial grid. Subsequently, we greedily select the positions based on the maximal $\Delta T_{\mathrm{mrt}}^t$ (TopK). During the iterative refinement, we perturb the current locally optimal tree position configurations using a genetic algorithm (Srinivas & Patnaik, 1994) (PerturbationWithGA). The initial population of the genetic algorithm comprises the current best (local) optima–we keep track of the five best optima–and randomly generated placements based on a sampling probability of

$$p_{\Delta T_{\mathrm{mrt}}^t} = \frac{\exp \Delta T_{\mathrm{mrt}_{i,j}}^t / \tau}{\sum\limits_{i,j} \exp \Delta T_{\mathrm{mrt}_{i,j}}^t / \tau} \quad , \tag{2}$$

where the temperature $\tau$ governs the entropy of $p_{\Delta T_{\mathrm{mrt}}^t}$. Subsequently, we refine all perturbed tree positions from the genetic algorithm with the hill climbing algorithm (HillClimbing), similar to Wallenberg et al. (2022). In particular, we repeatedly cycle over all trees of $s_*'$, try to move them within the adjacent eight neighbors, and accept the move if it improves $T_{\mathrm{mrt}}^{M,\phi}$. If the candidate $s_*'$ improves upon our current optima $S_*$, we accept and add it to our history of local optima $S_*$. Throughout the search, we ensure that trees are not placed on buildings nor water, and trees have no overlapping canopies. Algorithm 1 provides pseudocode.

**Theoretical analysis** It is easy to show that our optimization method finds the optimal tree placements given an unbounded number of iterations and sufficiently good $T_{\mathrm{mrt}}$ modeling.

**Lemma 1** ($p_{\Delta T_{\mathrm{mrt}_{i,j}}^t} > 0$). *The probability for all possible tree positions $(i,j)$ is $p_{\Delta T_{\mathrm{mrt}_{i,j}}^t} > 0$.*

*Proof.* Since the exponential function $\exp$ in Equation 2 is always positive, it follows that $p_{\Delta T_{\mathrm{mrt}_{i,j}}^t} > 0$ and the denominator is always non-zero. Thus, the probabilities are well-defined. $\square$

**Theorem 1** (Convergence to global optimum). *Our optimization method (Algorithm 1) converges to the globally optimal tree positions as (i) the number of iterations approaches infinity and (ii) the estimates of our $T_{mrt}$ modeling (Sections 3.2 and 3.3) are proportional to the true aggregated, point-wise $T_{\mathrm{mrt}}^{M,\phi}$ for an aggregation function $\phi$ and time period $M$.*

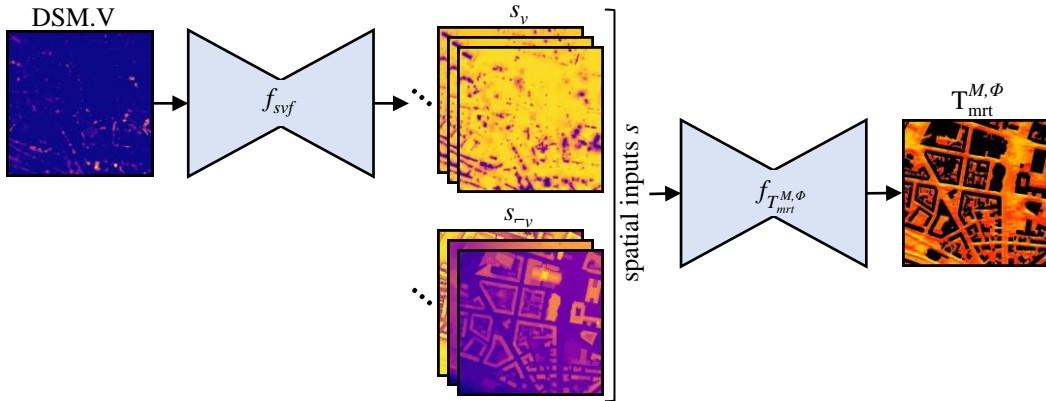

Figure 2: Overview of $T_{\mathrm{mrt}}^{M,\phi}$ modeling. To account for changes in vegetation during optimization, we modify the digital surface model for vegetation (DSM.V) and update depending spatial inputs (sky view factor maps for vegetation) with the model $f_{\mathrm{svf}}$. The model $f_{T_{\mathrm{mrt}}^{M,\phi}}$ takes these updated vegetation-related $s_v$ and non-vegetation-related spatial inputs $s_{\neg v}$ to estimate the aggregated, point-wise $T_{\mathrm{mrt}}^{M,\phi}$ for a given aggregation function $\phi$, e.g., mean, and time period $M$, e.g., heatwave event.

*Proof.* We are guaranteed to sample the globally optimal tree positions with an infinite budget (assumption (i)), as the perturbation step in our optimization method (`PerturbationWithGA`) randomly interleaves tree positions with positive probability (Lemma 1). Since our optimization method directly compares the effectiveness of tree positions using our $T_{\mathrm{mrt}}^{M,\phi}$ modeling pipeline–that yields estimates that are proportional to true $T_{\mathrm{mrt}}^{M,\phi}$ values (assumption (ii))–we will accept them throughout all steps of our optimization method and, consequently, find the global optimum. □

## 3.2 AGGREGATED, POINT-WISE MEAN RADIANT TEMPERATURE MODELING

Above optimization procedure is zero-order and, thus, requires fast evaluations of $T_{\mathrm{mrt}}$ to be computationally feasible. Recently, Briegel et al. (2023) employed a U-Net (Ronneberger et al., 2015) model $f_{T_{\mathrm{mrt}}}$ to estimate point-wise $T_{\mathrm{mrt}}$ for given spatial and meteorological inputs at a certain point in time. They trained the model on data generated by the microscale (building-resolving) SOLWEIG physical model (Lindberg et al., 2008) (refer to Appendix C for more details on SOLWEIG). However, our primary focus revolves around reducing aggregated, point-wise $T_{\mathrm{mrt}}^{M,\phi}$ for an aggregation function $\phi$, e.g., mean, and time period $M$, e.g., multiple days of a heatwave event. Thus, above approach would require the computation of point-wise $T_{\mathrm{mrt}}$ for all $|M|$ meteorological inputs of the time period $M$, followed by the aggregation with function $\phi$.[4] However, this procedure becomes prohibitively computationally expensive for large time periods.

To mitigate this computational bottleneck, we propose to learn a U-Net model

$$f_{T_{\mathrm{mrt}}^{M,\phi}}(\cdot) \approx \phi(\{f_{T_{\mathrm{mrt}}}(\cdot, m) \mid \forall m \in M\}) \tag{3}$$

that directly approximates aggregated, point-wise $T_{\mathrm{mrt}}^{M,\phi}$ for a given aggregation function $\phi$ and time period $M$. For training data, we computed aggregated, point-wise $T_{\mathrm{mrt}}^{M,\phi}$ for a specified aggregation function $\phi$ and time period $M$ with aforementioned (slow) procedure. However, note that this computation has to be done only once for the generation of training data. During inference, the computational complexity is effectively reduced by a factor of $\mathcal{O}(|M|)$.

## 3.3 MAPPING OF TREE PLACEMENTS TO THE SPATIAL INPUTS

During our optimization procedure (Section 3.1), we optimize the placement of trees by directly modifying the digital surface model for vegetation that represents the trees' canopies. However, depending spatial inputs (i.e., sky view factor maps for vegetation) cannot be directly modified and

---

[4]For sake of simplicity, we assumed that the spatial input is static over the entire time period.

conventional procedures are computationally expensive. Hence, we propose to estimate the sky view factor maps from the digital surface model for vegetation with another U-Net model $f_{\mathrm{svf}}$. To train this model $f_{\mathrm{svf}}$, we repurposed the conventionally computed sky view factor maps, that were already required for computing point-wise $T_{\mathrm{mrt}}$ with SOLWEIG (Section 3.2).

## 4 EXPERIMENTAL EVALUATION

In this section, we evaluate our optimization approach for tree placements across diverse study areas and time periods. We considered the following five study areas: `city-center` an old city-center, `new r.a.` a recently developed residential area (r.a.) where the majority of buildings were built in the last 5 years, `medium-age r.a.` a medium, primarily residential district built 25-35 years ago, `old r.a.` an old building district where the majority of buildings are older than 100 years, and `industrial` an industrial area. These areas vary considerably in their characteristics, e.g., existing amount of vegetation or proportion of sealed surfaces. Further, we considered the following time periods $M$: hottest `day` (and `week`) in 2020 based on the (average of) maximum daily air temperature, the entire `year` of 2020, and the `decade` from 2011 to 2020. While the first two time periods focus on the most extreme heat stress events, the latter two provide assessment over the course of longer time periods, including seasonal variations. We compared our approach with `random` (positioning based on random chance), `greedy` $T_{\mathrm{mrt}}$ (maximal $T_{\mathrm{mrt}}$), `greedy` $\Delta T_{\mathrm{mrt}}$ (maximal $\Delta T_{\mathrm{mrt}}$), and a `genetic` algorithm. We provide the hyperparameters of our optimization method in Appendix D. Model and training details for $T_{\mathrm{mrt}}$ and $T_{\mathrm{mrt}}^{M,\phi}$ estimation are provided in Appendix E. Throughout our experiments, we used the mean as aggregation function $\phi$. While all optimization algorithms used the faster direct estimation of aggregated, point-wise $T_{\mathrm{mrt}}^{M,\phi}$ with the model $f_{T_{\mathrm{mrt}}^{M,\phi}}$, we evaluated the final found tree placements by first predicting point-wise $T_{\mathrm{mrt}}$ for all $|M|$ meteorological inputs across the specified time period $M$ with the model $f_{T_{\mathrm{mrt}}}$ and subsequently aggregated these estimations. To quantitatively assess the efficacy of tree placements, we quantified the change in point-wise $T_{\mathrm{mrt}}$ ($\Delta T_{\mathrm{mrt}}$ [K]), averaged over the $500\,\mathrm{m}$ x $500\,\mathrm{m}$ study area ($\Delta T_{\mathrm{mrt}}$ area$^{-1}$ [Km$^{-2}$]), or averaged over the size of the canopy area ($\Delta T_{\mathrm{mrt}}$ canopy area$^{-1}$ [Km$^{-2}$]). We excluded building footprints and open water areas from our evaluation criteria. Throughout our experiments, we assumed that tree placements can be considered on both public and private property.

### 4.1 EVALUATION OF MEAN RADIANT TEMPERATURE MODELING

We first assessed the quality of our $T_{\mathrm{mrt}}$ and $T_{\mathrm{mrt}}^{M,\phi}$ modeling (Sections 3.2 and 3.3). Our model for estimating point-wise $T_{\mathrm{mrt}}$ ($f_{T_{\mathrm{mrt}}}$, Section 3.2) achieved a L1 error of $1.93\,\mathrm{K}$ compared to the point-wise $T_{\mathrm{mrt}}$ calculated by the physical model SOLWEIG (Lindberg et al., 2008). This regression performance is in line with Briegel et al. (2023) who reported a L1 error of $2.4\,\mathrm{K}$. Next, we assessed our proposed model $f_{T_{\mathrm{mrt}}^{M,\phi}}$ that estimates aggregated, point-wise $T_{\mathrm{mrt}}^{M,\phi}$ for aggregation function $\phi$ (i.e., mean) over a specified time period $M$ (Section 3.2). We found only a modest increase in L1 error by $0.46\,\mathrm{K}$ (for time period $M$=`day`), $0.42\,\mathrm{K}$ (`week`), $0.35\,\mathrm{K}$ (`year`), and $0.18\,\mathrm{K}$ (`decade`) compared to first predicting point-wise $T_{\mathrm{mrt}}$ for all $M$ meteorological inputs with model $f_{T_{\mathrm{mrt}}}$ and then aggregating them. While model $f_{T_{\mathrm{mrt}}^{M,\phi}}$ is slightly worse in regression performance, we want to emphasize its substantial computational speed-ups. To evaluate the computational speed-up, we used a single NVIDIA RTX 3090 GPU and averaged estimation times for $T_{\mathrm{mrt}}^{M,\phi}$ over five runs. We found computational speed-ups by up to 400,000x (for the time period `decade` with $|M| = 87,672$ meteorological inputs). Lastly, our estimation of sky view factors from the digital surface model for vegetation with model $f_{\mathrm{svf}}$ (Section 3.3) achieved a mere L1 error of $0.047\,\%$ when compared to conventionally computed sky view factor maps. Substituting the conventionally computed sky view factor maps with our estimates resulted in only a negligible regression performance decrease of ca. $0.2\,\mathrm{K}$ compared to SOLWEIG's estimates using the conventionally computed sky view factor maps.

### 4.2 EVALUATION OF OPTIMIZATION METHOD

We assessed our optimization method by searching for the positions of $k$ newly added trees. We considered uniform tree specimens with spherical crowns, tree height of $12\,\mathrm{m}$, canopy diameter of $9\,\mathrm{m}$, and trunk height of $25\,\%$ of the tree height (following the default settings of SOLWEIG).

Table 1: Quantitative results ($\Delta T_{\mathrm{mrt}}$ area$^{-1}$ [Km$^{-2}$] ± standard error) for positioning 50 added trees of a height of $12\,\mathrm{m}$ and canopy diameter of $9\,\mathrm{m}$, yielding an additional canopy area size of $4050\,\mathrm{m}^2$ ($1.62\,\%$ of each area), averaged over the five study areas.

| Method | day | week | year | decade |
|---|---|---|---|---|
| random | -0.1156 ± 0.007 | -0.0891 ± 0.0063 | 0.0179 ± 0.0025 | 0.0208 ± 0.0024 |
| greedy $T_{\mathrm{mrt}}$ | -0.1825 ± 0.0088 | -0.1219 ± 0.0034 | 0.016 ± 0.0014 | 0.0176 ± 0.0014 |
| greedy $\Delta T_{\mathrm{mrt}}$ | -0.2248 ± 0.0094 | -0.1791 ± 0.0065 | -0.0206 ± 0.0032 | -0.0212 ± 0.0053 |
| genetic | -0.2585 ± 0.0108 | -0.1927 ± 0.009 | -0.0172 ± 0.0048 | -0.0228 ± 0.0055 |
| ILS$^{\dagger}$ (ours) | **-0.2996** ± 0.0113 | **-0.2331** ± 0.0083 | **-0.0309** ± 0.0045 | **-0.0335** ± 0.0065 |

$^{\dagger}$: ILS = iterated local search.

Table 2: Ablation study over different choices of our optimization method for the time period week averaged across the five study areas for 50 added trees of height of $12\,\mathrm{m}$ and crown diameter of $9\,\mathrm{m}$.

| TopK | PerturbationWithGA | HillClimbing | Iterations | $\Delta T_{\mathrm{mrt}}$ area$^{-1}$ [Km$^{-2}$] |
|---|---|---|---|---|
| ✓ | - | - | - | -0.1793 |
| - | ✓ | ✓ | ✓ | -0.1955 |
| ✓ | - | ✓ | ✓ | -0.2094 |
| ✓ | ✓ | - | ✓ | -0.2337 |
| ✓ | ✓ | ✓ | - | -0.2302 |
| ✓ | ✓ | ✓ | ✓ | **-0.2345** |

**Results** Figure 1 illustrates the efficacy of our approach in reducing point-wise $T_{\mathrm{mrt}}$ across diverse urban districts and time periods. We observe that trees predominantly assume positions on east-to-west aligned streets and large, often paved spaces. However, tree placement becomes more challenging with longer time scales. This observation is intricately linked to seasonal variations, as revealed by our analyses in Section 4.3. In essence, the influence of trees on $T_{\mathrm{mrt}}$ exhibits a duality–contributing to reductions in summer and conversely causing increases in winter. Furthermore, this dynamic also accounts for the observed variations in $T_{\mathrm{mrt}}$ on the northern and southern sides of the trees, where decreases and increases are respectively evident. Table 1 affirms that our optimization method consistently finds better tree positions when compared against the considered baselines.

**Ablation study** We conducted an ablation study by selectively ablating components of our optimization method. Specifically, we studied the contributions of the greedy initialization strategy (TopK) by substituting it with random initialization, as well as (de)activating perturbation (PerturbationWithGA), local search (HillClimbing), or the iterative design (Iterations). Table 2 shows the positive effect of each component. It is noteworthy that the iterated design may exhibit a relatively diminished impact in scenarios where the greedy initialization or first iteration already yield good or even the (globally) optimal tree positions.

## 4.3 ANALYSES

Given the found tree placements from our experiments in Section 4.2, we conducted analyses on various aspects (daily variation, seasonal variation, number of trees, tree geometry variation). Figure 3 shows a noteworthy duality caused by daily and seasonal variations. Specifically, trees exert a dual influence, reducing $T_{\mathrm{mrt}}$ during daytime and summer season, while conversely increasing it during nighttime and winter season. To understand the impact of meteorological parameters on this, we trained an XGBoost classifier (Chen et al., 2015) on each study area and all meteorological inputs from 2020 (year) to predict whether the additional trees reduce or increase $T_{\mathrm{mrt}}$. We assessed feature importance using SHAP (Shapley, 1953; Lundberg & Lee, 2017) and found that incoming shortwave radiation $I_g$ emerges as the most influential meteorological parameter. Remarkably, a

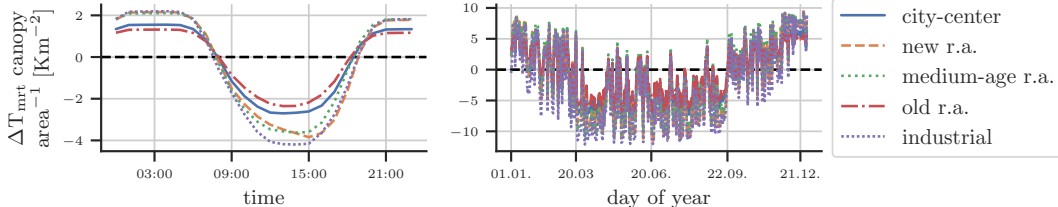

Figure 3: Daily (left) and seasonal variation (right) reduce $T_{mrt}$ during daytime and summer season, while conversely increase it during nighttime and winter season. Results based on experiments adding 50 trees, each with a height of $12\,m$ and a crown diameter of $9\,m$, for the time period `year`.

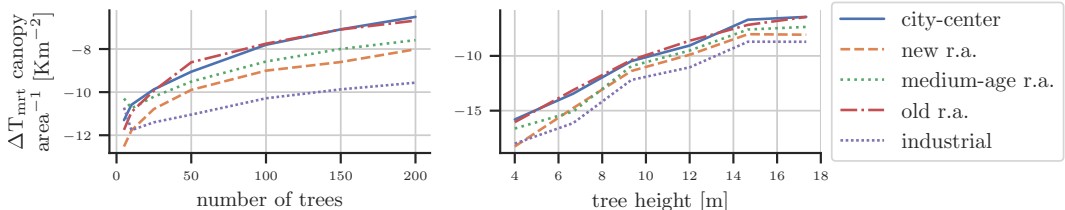

Figure 4: Increasing the number of trees (left) and tree height (right) has diminishing returns for the reduction of $T_{mrt}$. Results are based on the experiment adding trees for the time period `week`.

simple classifier of the form

$$y = \begin{cases} T_{mrt} \text{ decreases,} & I_g > 96\,\text{Wm}^{-2} \\ T_{mrt} \text{ increases,} & \text{otherwise} \end{cases}, \tag{4}$$

achieves an average accuracy of $97.9\,\% \pm 0.005\,\%$, highlighting its predictive prowess.

Besides the above, Figure 4 reveals a pattern of diminishing returns as we increase the extent of canopy cover, achieved either by adding more trees or by using larger trees. This trend suggests that there may be a point of saturation beyond which achieving further reductions in $T_{mrt}$ becomes progressively more challenging. To corroborate this trend quantitatively, we computed Spearman rank correlations between $\Delta T_{mrt}$ canopy area$^{-1}$ and the size of the canopy area; also including pre-existing trees with a minimum height of $3\,m$. We found high Spearman rank correlations of $0.72$ or $0.73$ for varying number of trees or tree heights, respectively. Notwithstanding the presence of diminishing returns, we still emphasize that each tree leads to a palpable decrease in $T_{mrt}$, thereby enhancing outdoor human thermal comfort–an observation that remains steadfast despite these trends.

### 4.4 COUNTERFACTUAL PLACEMENT OF TREES

In our previous experiments, we always added trees to the existing urban vegetation. However, it remains uncertain whether the placement of existing trees, determined by natural evolution or human-made planning, represents an optimal spatial arrangement of trees. Thus, we pose the counterfactual question (Pearl, 2009): *could alternative tree positions have retrospectively yielded reduced amounts of heat stress*? To answer this counterfactual question, we identified all existing trees from the digital surface model for vegetation with a simple procedure based on the watershed algorithm (Soille & Ansoult, 1990; Beucher & Meyer, 2018)–which is optimal in identifying non-overlapping trees, i.e., the maximum point of the tree does not overlap with any other tree, with strictly increasing canopies towards each maximum point–and optimized their placements for the hottest `week` in 2020 (heat-wave condition). We only considered vegetation of a minimum height of $3\,m$ and ensured that the post-extraction size of the canopy area does not exceed the size of the (f)actual canopy area.

**Results** We found alternative tree placements that would have led to a substantial reduction of $T_{mrt}$ by an average of $0.83\,K$. Furthermore, it would have resulted in a substantial reduction of hours with $T_{mrt}$ exceeding $60\,°C$–a recognized threshold for heat stress (Lee et al., 2013; Thorsson et al., 2017)–by on average $19.7\,\%$ throughout the duration of the heatwave event (`week`). This strongly suggests that the existing placements of trees may not be fully harnessed to their optimal capacity. Notably, the improvement by relocation of existing trees is significantly larger than the effect of

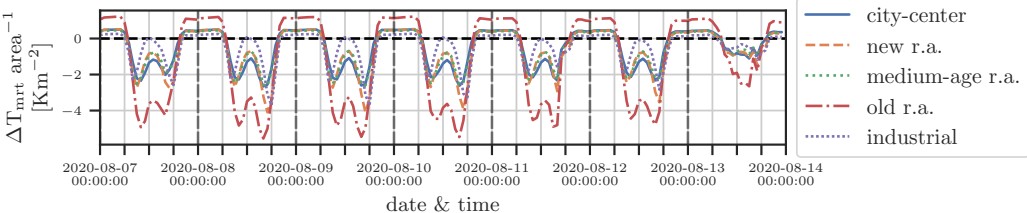

Figure 5: Alternative placements of existing trees substantially reduces $T_{\mathrm{mrt}}$ during daytime. Optimization ran for the hottest `week` in 2020 (heatwave condition).

50 added trees (0.23 K; see Table 1). Figure 5 visualizes the change in $T_{\mathrm{mrt}}$ across each hour of the hottest `week` in 2020. Intriguingly, they reveal peaks during morning and afternoon hours. By inspecting the relocations of trees (see Figure 10), we found that trees tend to be relocated from spaces with already ample shading from tree canopies and buildings to large, open, typically sealed spaces without trees, such as sealed plazas or parking lots.

## 5 LIMITATIONS

The main limitation, or strength, of our approach is assumption (ii) from Theorem 1 that the model $f_{T_{\mathrm{mrt}}^{M,\phi}}$ yields estimates for that are (at least) proportional to the true aggretated, point-wise $T_{\mathrm{mrt}}^{M,\phi}$ for aggregation function $\phi$ and time period $M$. Our experimental evaluation affirms the viability of this approximation, but it remains an assumption. Another limitation is that we assumed a static urban environment, contrasting the dynamic real world. Further, we acknowledge the uniform tree parameterization, i.e., same tree geometry, species, or transmissivity. While varying tree geometry could be explored further in future works, both latter are limitations of SOLWEIG, which we rely on to train our models. In a similar vein, our experiments focused on a single city, which may not fully encompass the diversity of cities worldwide. We believe that easier data acquisition of spatial input data, e.g., through advances in canopy and building height estimation (Lindemann-Matthies & Brieger, 2016; Tolan et al., 2023), could facilitate the adoption of our approach to other cities. Further, our experiments lack a distinction between public and private property, as well as does not incorporate considerations regarding the actual ecological and regulatory feasibility of tree positions, e.g., trees may be placed in the middle of streets. Lastly, our approach does not consider the actual zones of activity and pathways of pedestrians. Future work could address these limitations by incorporating comprehensive data regarding the feasibility, cost of tree placements and pedestrian pathways, with insights from, e.g., urban forestry or legal experts, as well as considering the point-wise likelihood of humans sojourning at a certain location. Finally, other factors, such as wind, air temperature, and humidity, also influence human thermal comfort, however vary less distinctly spatially and leave the integration of such for future work.

## 6 CONCLUSION

We presented a simple and scalable method to optimize tree locations across large urban areas and time scales to mitigate pedestrian-level heat stress by optimizing human thermal comfort expressed by $T_{\mathrm{mrt}}$. We proposed a novel approach to efficiently model aggregated, point-wise $T_{\mathrm{mrt}}^{M,\phi}$ for a specified aggregation function and time period, and optimized tree placements through an instantiation of the iterated local search framework with tailored adaptations. Our experimental results corroborate the efficacy of our approach. Interestingly, we found that the existing tree stock is not harnessed to its optimal capacity. Furthermore, we unveiled nuanced temporal effects, with trees exhibiting distinct decreasing or increasing effects on $T_{\mathrm{mrt}}$ during day- and nighttime, as well as across summer and winter season. Future work could scale our experiments to entire cities, explore different aggregation functions e.g., top 5 % of the most extreme heat events, integrate density maps of pedestrians, or optimize other spatial inputs, e.g., land cover usage.

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

# A    MEAN RADIANT TEMPERATURE

The mean radiant temperature $T_{\text{mrt}}$ [°C] is a driving meteorological parameter for assessing the radiation load on humans. During the day, it is of particular importance in determining human outdoor thermal comfort. $T_{\text{mrt}}$ varies spatially, e.g., standing in direct sunlight on a hot day results in a less favorable thermal experience for the human body than seeking shelter in shaded areas. $T_{\text{mrt}}$ is defined as the "uniform temperature of an imaginary enclosure in which radiant heat transfer from the human body equals the radiant heat transfer in the actual non-uniform enclosure" by ASHRAE (2001). That is, $T_{\text{mrt}}$ can be calculated by measured values of surrounding objects and their position w.r.t. the person. Formally, $T_{\text{mrt}}$ can be computed by

$$\text{T}_{\text{mrt}}^4 = \sum_{i=1}^{N} T_i^4 F_{p-i} \qquad , \tag{5}$$

where $T_i$ is the surface temperature of the $i$-th surface and $F_{p-i}$ is the angular factor between a person and the $i$-th surface (ASHRAE, 2001). Alternatively, we can use the six-directional approach of Höppe (1992) through estimation of short- and longwave radiation fluxes of six directions (upward, downward, and the four cardinal directions), as follows:

$$\text{T}_{\text{mrt}} = \frac{0.08(T_p^{\text{up}} + T_p^{\text{down}}) + 0.23(T_p^{\text{left}} + T_p^{\text{right}}) + 0.35(T_p^{\text{front}} + T_p^{\text{back}})}{2(0.08 + 0.23 + 0.35)} \qquad , \tag{6}$$

where $T_{pr}$ is the plane radiant temperature (Korsgaard, 1949).

# B    SPATIAL AND METEOROLOGICAL INPUT DATA

To predict point-wise $T_{\text{mrt}}$ we use the following spatial inputs:

- Digital elevation model [m]: representation of elevation data of terrain excluding surface objects.
- Digital surface model with heights of ground and buildings [m]: heights of ground and buildings above sea level.
- Digital surface model with heights of vegetation [m]: heights of vegetation above ground level.
- Land cover class map [{paved, building, grass, bare soil, water}]: specifies the land-usage.
- Wall aspect [°]: aspect of walls where a north-facing wall has a value of zero.
- Wall height [m]: specifies the height of a wall of a building.
- Sky view factor maps [%]: cosine-corrected proportion of the visible sky hemisphere from a specific location from earth's surface by the total solid angle of the entire sky hemisphere.

Figure 6 shows exemplar spatial inputs and Table 3 provides exemplar temporal (meteorological) inputs. Note that the model $f_{T_{\text{mrt}}}$ requires both spatial and temporal (meteorological) inputs, whereas our proposed model $f_{T_{\text{mrt}}^{M,\phi}}$ only requires spatial inputs, as it directly outputs aggregated, point-wise $T_{\text{mrt}}^{M,\phi}$ for a specified aggregation function $\phi$ and time period $M$ with its respective meteorological inputs.

The data comprises of a total of 61 areas, each spanning an area of $500\,\text{m}$ x $500\,\text{m}$. Each area is characterized by its spatial inputs (see above). Meteorological input data was provied by an urban weather station located in the northern part of the city of CITY. For more details on the data acquisition as well as pre-processing of data, refer to Briegel et al. (2023). The mean radiation temperature was computed for 68 days with hourly resolution with SOLWEIG (Lindberg et al., 2008) and provided by Briegel et al. (2023). We divided the data into training and test data by area, i.e., the five areas to be optimized constituted the test set and the others constituted the training set. We used this training-test split throughout all of our experiments.

We used this data to train $f_{T_{\text{mrt}}}$ following Briegel et al. (2023). To train $f_{T_{\text{mrt}}^{M,\phi}}$, we computed aggregated, point-wise $T_{\text{mrt}}^{M,\phi}$ with $f_{T_{\text{mrt}}}$, i.e., we computed $T_{\text{mrt}}$ for each time step and aggregated them

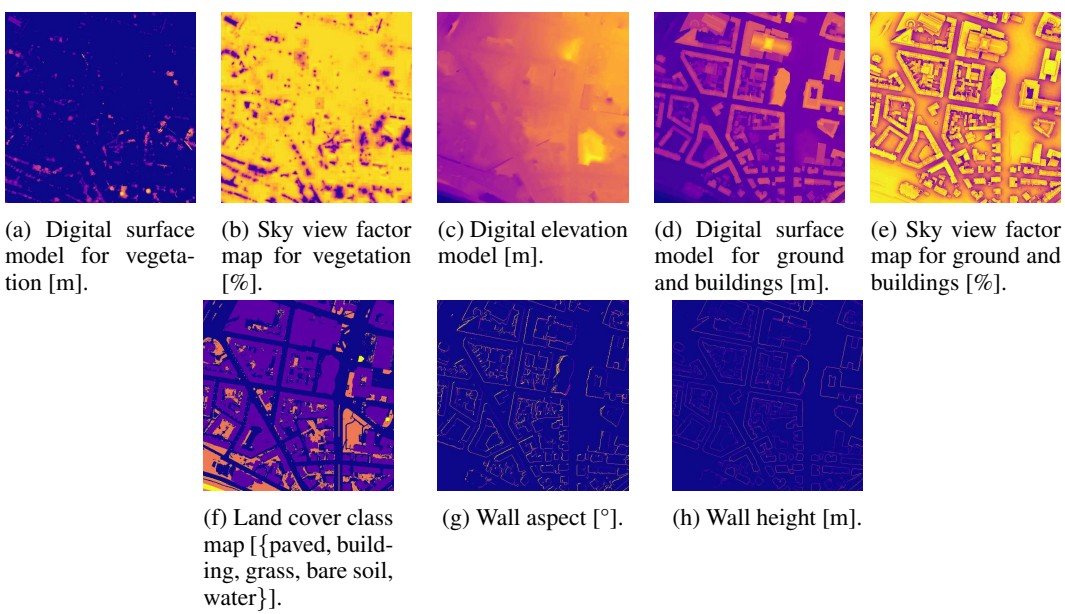

(a) Digital surface model for vegetation [m].

(b) Sky view factor map for vegetation [%].

(c) Digital elevation model [m].

(d) Digital surface model for ground and buildings [m].

(e) Sky view factor map for ground and buildings [%].

(f) Land cover class map [{paved, building, grass, bare soil, water}].

(g) Wall aspect [°].

(h) Wall height [m].

Figure 6: Exemplar spatial inputs. Note that we omit sky view factor maps for vegetation or ground and building for the four cardinal directions (north, east, south, west) for visualization.

Table 3: Exemplar meteorological inputs.

| Date & time | Air temperature [°C] | Wind speed [ms⁻¹] | Wind direction [°] | Incoming shortwave radiation [Wm⁻²] | Precipitation [mm] | Relative humidity [%] | Barometric pressure [kPA] | Elevation angle [°] | Azimuth angle [°] |
|---|---|---|---|---|---|---|---|---|---|
| 2020-01-01 00:00:00 | -1.31 | 1.97 | 107.72 | 0.0 | 0.0 | 75.88 | 976.4 | 0.0 | 343.009 |
| 2020-03-20 06:00:00 | 9.07 | 0.64 | 114.71 | 0.0 | 0.0 | 83.46 | 964.7 | 0.0 | 83.264 |
| 2020-06-20 11:00:00 | 18.43 | 1.15 | 314.46 | 652.27 | 0.02 | 68.04 | 966.2 | 59.601 | 135.899 |
| 2020-09-22 16:00:00 | 22.35 | 1.43 | 245.3 | 310.98 | 0.04 | 56.1 | 955.1 | 22.754 | 242.247 |
| 2020-12-21 21:00:00 | 7.19 | 4.25 | 202.69 | 0.0 | 0.51 | 87.82 | 963.0 | 0.0 | 282.195 |

with the aggregation function $\phi$. This reduced the data to 61 and 5 for the training or test set, respectively. To train $f_{\text{svf}}$, we repurposed the sky view factor maps that were already required for the computation of $T_{\text{mrt}}$ with SOLWEIG. Here, we masked out the five test areas for training and used the remainder for training. Appendix E provides all training details of each model.

## C  MEAN RADIANT TEMPERATURE MODELING WITH SOLWEIG

SOLWEIG (Lindberg et al., 2008) is a popular method to estimate mean radiant temperature. It uses spatial and temporal (meteorological) inputs to model $T_{\text{mrt}}$ for a height of $1.1\,\text{m}$ of a standing or walking rotationally symmetric person using the six-dimension approach presented in Appendix A. We used the default model parameters:

- Emissivity ground: 0.95.
- Emissivity walls: 0.9.
- Albedo ground: 0.15.
- Albedo walls: 0.2.
- Transmissivity: $3\,\%$.
- Trunk height: $25\,\%$ of tree height.

Below, we provide a list of limitations within the physical modeling with SOLWEIG, which our approach consequently inherits, to make machine learning practitioners aware of these limitations:

- Static environment assumption: SOLWEIG assumes that the environment stays static. That is, trees do not grow, infrastructure does not change nor seasonal affects are accounted for.

- No distinction between different types of trees.

- No consideration of other important factors, such as wind, air temperature, and humidity.

Note that our approach inherits these limitations. Future progress in incorporating these in physical modeling would also be reflected in our approach.

## D HYPERPARAMETER CHOICES OF OPTIMZATION

We implemented the genetic algorithm with PyGAD[5]. Throughout our experiments, we used a population size of 20 with steady-state selection for parents, random mutation and single-point crossover. We used the current best optima (up to five) and random samples for the initial population. We set the temperature $\tau$ of Equation 2 to 1. We kept the best solution throughout the evolution. We used 1000 iterations within our optimization method. For the baseline `genetic` algorithm, we used 5000 iterations to account for larger compute due to the iterative design of our optimization approach.

For the `HillClimbing` algorithm, we adopted the design by Wallenberg et al. (2022). That is, we repeatedly cycle over all trees and try to move them within the adjacent eight neighbors. We accept the move if it improves upon the current aggregated, point-wise $T_{\mathrm{mrt}}^{M,\phi}$. We repeat this process until no further improvement can be found.

Lastly, we used five iterations within our iterated local search. We found this resulted in a good trade-off between the efficacy of the final tree placements and total runtime.

## E MODEL AND TRAINING DETAILS

**Model details**    We adopted the U-Net architecture from Briegel et al. (2023). Specifically, the models $f_{T_{\mathrm{mrt}}}$ and $f_{T_{\mathrm{mrt}}^{M,\phi}}$ receive inputs of size $16 \times h \times w$ and predict $T_{\mathrm{mrt}}$ or $T_{\mathrm{mrt}}^{M,\phi}$, respectively, of size of $h \times w$, where $h$ and $w$ are the height and width of the spatial input, respectively. The model $f_{\mathrm{svf}}$ receives an input of size $h \times w$ (digital surface model for vegetation) and outputs the sky view factor maps for vegetation of size of $5 \times h \times w$. All models use the U-Net architecture (Ronneberger et al., 2015) with a depth of 3 and base dimensionality of 64. Each stage of the encoder and decoder consist of a convolution or transposed convolution, respectively, followed by batch normalization (Ioffe & Szegedy, 2015) and ReLU non-linearity.

**Training details**    In the following, we provide the specific training details of all models.

- $f_{T_{\mathrm{mrt}}}$: We trained the model with L1 loss function for ten epochs using the Adam optimizer (Kingma & Ba, 2015) with learning rate of 0.001 and exponential learning rate decay schedule. We randomly cropped (256x256) the inputs during training.

- $f_{T_{\mathrm{mrt}}^{M,\phi}}$: We trained the model with L1 loss function for 5000 epochs and batch size of 32 with Adam optimizer (Kingma & Ba, 2015) with learning rate of 0.001 and exponential decay learning rate schedule. We randomly cropped (256x256) the inputs during training.

- $f_{\mathrm{svf}}$: We trained the model with L1 loss function for 20 epochs with Adam optimizer (Kingma & Ba, 2015) with learning rate of 0.001 and exponential decay learning rate schedule. We randomly cropped (256x256) the inputs during training.

Note that we have not tuned hyperparamerters and improved modeling performance could be achieved through hyperparameter optimization. However, since the goal of this work is provide a case study how deep learning methods can effectively be used to adapt to climate change, we omitted hyperparameter optimization.

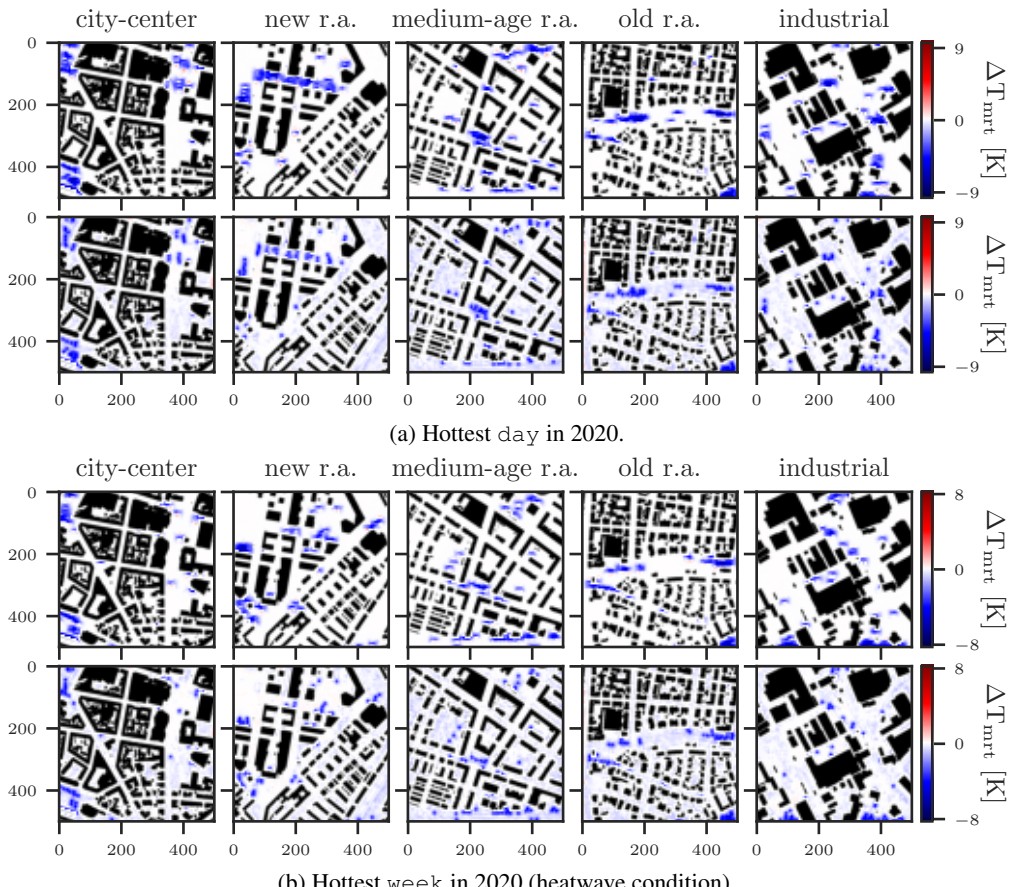

Figure 7: Evaluation of aggregated results with $f_{T\mathrm{mrt}}$ (top row of subfigures), i.e., predicting $T_{\mathrm{mrt}}$ for each point in time and then aggregating the results, yields comparable results to evaluating with SOLWEIG (bottom row of the subfigures).

## F SUPPLEMENTARY EXPERIMENTAL RESULTS

### F.1 COMPARISON OF OUR EVALUATION PROTOCOL TO SOLWEIG

We compared our evaluation protocol, i.e., predicting $T_{\mathrm{mrt}}$ for each time step with $f_{T\mathrm{mrt}}$ and then aggregating results with the aggregation function $\phi$, to running evaluation with SOLWEIG instead. Figure 7 compares the results for the time periods day and week. Note that the other time periods, i.e., year and decade, are prohibitively expensive to evaluate with SOLWEIG. We find that our evaluation protocol closely matches SOLWEIG's results, i.e., L1 errors of 0.296 K and 0.25 K for day or week, respectively. This affirms the validity of our evaluation strategy.

### F.2 TREE PLACEMENTS DURING SUMMER VS. WINTER SEASON

Figure 8 compares the tree placements during summer and winter season. We observe that trees are predominately positioned at large, open areas and clustered together during summer season, whereas they are more broadly spread over the neighborhood during winter season.

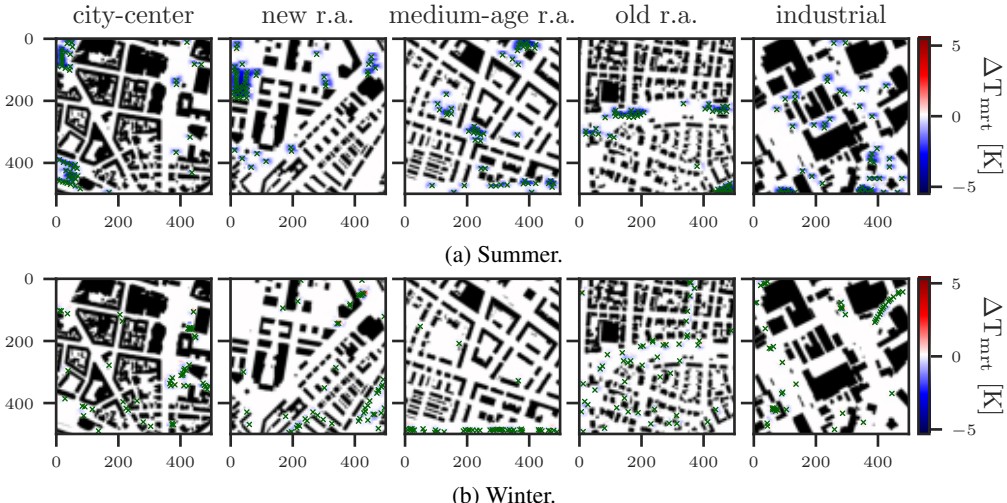

(a) Summer.

(b) Winter.

Figure 8: Comparison of tree placements between summer and winter season.

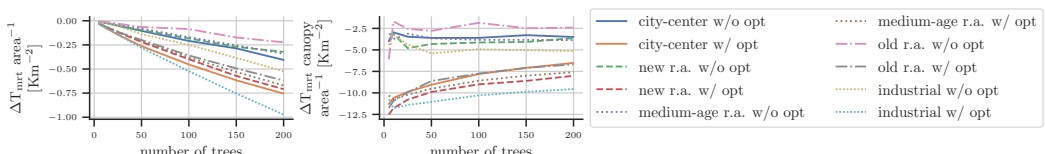

Figure 9: Optimization of tree positions yields significantly more reduction in $T_{\text{mrt}}$. It is especially pronounced for fewer number of trees. This shows the importance of optimizing the the positions of trees.

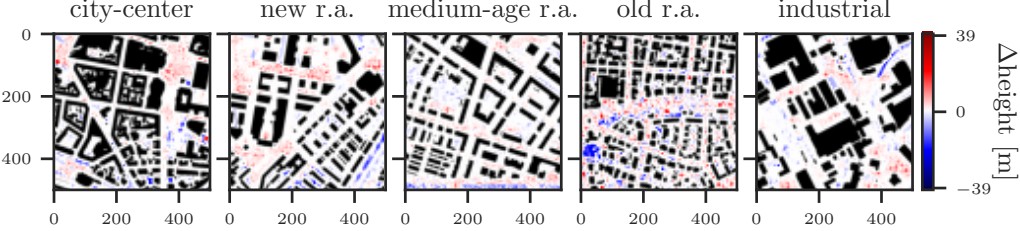

Figure 10: The alternative tree placements (Section 4.4) often move from north-to-south streets to west-to-east streets as well as open, typically paved spaces. Red indicates that vegetation (i.e., tree) was added, whereas blue indicates that vegetation was removed.

### F.3 IMPACT OF THE NUMBER OF TREES VS. OPTIMIZATION

Figure 9 shows the importance of the right positioning of trees through the optimization in comparison with the number of trees. It is therefore not only important to plant trees, but also to consider at which location they can be used most effectively.

### F.4 ALTERNATIVE PLACEMENTS IN THE COUNTERFACTUAL SCENARIO

Figure 10 visualizes the change in vegetation for our experiments from Section 4.4. We observe that trees were moved from north-to-south to west-to-east aligned streets as well as to large open, typically sealed spaces, such as sealed plazas or parking lots.

---

[5]https://github.com/ahmedfgad/GeneticAlgorithmPython, License: BSD-3

## G  BROADER IMPACT

We believe that our approach can empower urban decision-makers selecting effective measures for climate-sensitive urban planning and climate adapation, reduces power consumption, and democratizes access to planning tools to smaller communities as well as citizens. However, our approach could also be used improperly for urban planning by ignoring other important factors, such as the influence of trees on wind patterns, heavy rain events, or legal requirements. Moreover, adverse individuals may manipulate results to further their personal goals, e.g., they do not want trees in front of their homes, which may not necessarily align with societal goals.

**Carbon footprint estimate**  All components of our optimization approach cause carbon dioxide emissions. We estimated the emissions for our final experiments (including training of models, optimization, ablations etc.) using the calculator by Lacoste et al. (2019).[6] Experiments were conducted using an internal infrastructure, which has a carbon efficiency of $0.436\,\mathrm{kgCO_2eq/kWh}$.[7] A cumulative of ca. $380\,\mathrm{h}$ of computation was performed on various GPU hardware. Emissions are estimated to be ca. $44\,\mathrm{kgCO_2eq}$. Note that the actual carbon emissions over the course of this project is multiple times larger.

---

[6]https://mlco2.github.io/impact/
[7]The global carbon efficiency figure will be replaced by that of the country from 2022 upon acceptance. Note that carbon efficiency figures for 2023 were not released at time of writing.

