# OpenReview forum: "Climate-sensitive Urban Planning through Optimization of Tree Placements"
_ICLR.cc/2024/Conference — Submitted to ICLR 2024_

### Official Review · Reviewer_oMtG · 2023-10-22

**Soundness:** 3 good
**Presentation:** 3 good
**Contribution:** 2 fair
**Rating:** 5
**Confidence:** 3

**Summary:**

This paper discusses a spatial optimization problem - how a certain number of items (in this case trees) can be distributed over a spatial grid system to optimize a particular objective (in this case Mean Radiant Temperature-MRT). For this task, they discuss a genetic algorithm based approach to identify the optimal distribution. A neural network (U-Net) is used to estimate the impact of placement of each tree on aggregate MRT. This U-Net is used to approximate a process-based model that estimates point-wise MRT based on meteorological and vegetation features. A counterfactual-based analysis is also carried out, where optimal placement of even existing trees are considered in addition to new ones.

**Strengths:**

1) The paper considers a novel spatial optimization problem with a very relevant application - climate-sensitive urban vegetation planning.
2) The paper develops a neural network surrogate for an existing process-based model (Solweig), this can be an useful contribution to the growing subfield of ML-based surrogates for process models

**Weaknesses:**

1) No technical contribution in ML. Even the considered surrogate model is a simple U-Net. It is not used for anything other than approximation of Solweig. If it gives computational benefits, this aspect is never discussed, nor is the quality of approximation evaluated
2) The optimization seems to be done one tree at a time, rather than jointly over an entire configuration. This need not be a weakness, but the approaches need to be compared
3) The problem is solved in rather artificial settings, without considering many realistic effects (as mentioned by the authors themselves, in the limitations section)
4) The proposed approach seems to be too specific to one application to be of sufficient interest to the ICLR audience in general

**Questions:**

1) Algorithm 1 suggests that the MRT seems to have been pre-computed for placing each tree separately. But in each iteration of the GA, the existing tree placements are partially perturbed. Do we not need to recompute the MRT once again for each perturbation?
2) Is U-Net the best model for generating the spatial maps? Have different architectures been compared? I didn't see much discussion about this
3) Several limitations have been mentioned in the paper itself. Can those be overcome within the proposed framework itself, or do they need completely different approaches?

---

> ### Author Response · Authors · 2023-11-19
> **Re: Official Review of Submission3376 by Reviewer oMtG**
>
> We thank the reviewer for taking the time to review our paper and raising interesting points. We are glad that the reviewer acknowledges that our work “considers a novel spatial optimization problem with a very relevant application”. Below, we address the concerns and questions of the reviewer.
>
> ### W1,W4: ML contribution & missing evaluation
>
> We kindly refer the reviewer to Section 4.1 that thoroughly compares our method to SOLWEIG. Next, we provide further details about the computational speed-ups of our method: we report speed-ups of factors of 30, 17, 44000, and 400000 for the time periods: day, week, year, or decade, respectively, when comparing using our model that directly outputs the aggregated mean radiant temperature to first predicting the mean radiant temperature for each time step and then aggregating over all time steps. Note that using deep learning methods compared to physical modeling with SOLWEIG achieves another factor of ca. 22x [1]. Note that prior works exclusively used physical simulation models. To contextualize the amount of speed-up we gain, let us consider an example: consider that the optimization takes 1 minute with our method and optimizes the tree positions for the time period decade. Consequently, the estimated runtime for SOLWEIG would be 1 min\*400000\*22=ca. 16.73 years.
>
> As mentioned by the reviewer, the main contribution of our work is the introduction of a “novel spatial optimization problem with a very relevant application”. Prior work mostly focused just on computational efficiency or improved modeling, i.e., improving the forward direction. In contrast, we consider the backward direction, i.e., making interventions in the input to optimize some physical quantity, i.e., in our case reducing the mean radiant temperature. Besides that we do *not* use it as a surrogate model but learn to directly predict aggregated results which goes beyond a simple replacement.
>
> ### W2, Q1: Sequential or joint optimization
>
> After greedily placing trees subsequently, the iterative search jointly optimizes all tree positions. We note that sequential placements would incur significantly more computational costs and, thus, we did not run such.
>
> Thanks for pointing out the missing evaluation in the algorithm. We indeed need to evaluate the new candidate. We fixed the algorithm in the revised manuscript.
>
> ### W3, Q3: Simplifying assumptions
>
> We note that many limitations could be overcome in future work. The main issue for most limitations is the data bottleneck, i.e., the data often is not available, or limitations of physical modeling (refer to Appendix C for more details in the revised manuscript). Both could be overcome by future advances in easier access, e.g., by means of estimating them through satellite imagery, as well as advances in physical modeling.
>
> ### Q2: Model architecture choice
>
> Our approach is not specific to the model architecture. The goal of our work is not just to show good numbers but to outline a novel optimization problem that uses deep learning approaches for the optimization of physical quantities. Consequently, the model architecture is a non-important implementation detail in our opinion and future work could improve upon it.
>
> ---
>
> [1] Briegel, Ferdinand, et al. "Modelling long-term thermal comfort conditions in urban environments using a deep convolutional encoder-decoder as a computational shortcut." Urban Climate 2023.

---

> > ### Comment · Reviewer_oMtG · 2023-11-21
> >
> > I thank the authors for their responses. Some of my specific queries have been answered, hence I will slightly improve my rating. However, my primary concern about this work remains that the technical contributions and experimental evaluations are not suitable for ICLR. A better venue for this paper would be a journal on environmental data science or urban planning.

---

> ### Author Response · Authors · 2023-11-23
> **Thank you!**
>
> Thank you for the follow-up and updated score! We appreciated the discussion and your constructive feedback.
>
> Concerning the venue, we believe that ICLR is a good platform for our work. Our aim is to prompt ML experts leveraging their skills across various scientific domains to reconsider the prevailing emphasis mostly exclusively on computational efficiency or general forward modeling. Instead, we would be happy if the takeaway of our paper for them would be to consider alternative ML applications in the scientific domain more often, such as optimization of some physical quantity.

---

### Official Review · Reviewer_TK26 · 2023-10-31

**Soundness:** 3 good
**Presentation:** 2 fair
**Contribution:** 1 poor
**Rating:** 3
**Confidence:** 3

**Summary:**

To improve urban planning and reducing temperature stress from inhabitants, a U-Net is trained to predict mean radiant temperature from various meteorological and surface inputs, including vegetation. This vegetation input is optimized with an evolutionary algorithm to reveal ideal tree positions that minimize mean radiant temperature. The model runs several orders of magnitudes faster than conventional models but produces slightly deteriorated results.

**Strengths:**

_Originality:_ This work does not seem to be of particular originality in terms of ML methods. Hence, the manuscript might be better suited for a journal about application of ML to environmental processes. I do like the topic of research and see its importance, though, and would like to encourage the authors to submit the work to an according journal.

_Quality:_ The manuscript is sound. Results of the experiments support the claims.

_Clarity:_ I had some difficulties in understanding what kind of data was used in this work (see questions below).

_Significance:_ Even though the results are important, they do not seem to bring new insights, nor do they seem to be of particular novelty. Replacing physical models with surrogate ML methods has been demonstrated widely to result in substantial speed-ups.

_Further comments_:
- The section about limitations is well structured and captures relevant aspects. Most concerning to me is the restriction to a single study area, which means that results cannot be expected to hold for different climatic regions. Extending the experiments to different cities would be of great value for the manuscript.

**Weaknesses:**

1. Unclear what data has been used. Does the CityGML data come from [this](https://www.ogc.org/standard/CityGML/) homepage and is this all simulation data, or real observations? As far as I know, ERA5 data is only available on 30m resolution; how do you get to 1m resolution? Where do your digital elevation and surface models, your land cover, wall aspects and height, and sky view factor maps come from?
2. Benchmarking traditional physical model would be highly appreciated to understand the performance of your U-Net in terms of accuracy. That is, does your method incur a trade-off between accuracy and efficiency, and, in particular, how accurate is your U-Net to approximate $T_{mrt}$?
3. Is SOLVEIG a traditional model and do you benchmark this? Please help readers by clarifying this more thoroughly. What does an `L1 error of 1.93K` mean, is it computed between U-Net output and ground truth, or between U-Net output and SOLVEIG output?
5. Given that the solar incoming radiation $I_g$ has the biggest impact, how does the tree position make a difference? An ablation comparing the effect on $T_{mrt}$ caused by number of trees vs. the positioning of the trees would be interesting. Also, seing Figure 4, I do not quite agree that the number of trees saturates (although it certainly will at some point). In short: Does the number of trees or their position contribute more to a better $T_{mrt}$? (I have seen your comment `Notably, the improvement by relocation of existing trees is significantly larger than the effect of 50 added trees [...]`, but I could not quite understand the quantitative effect of each treatment, i.e., #trees vs position).

**Questions:**

1. In Figure 1: What does the legend show, is it difference in $T_{mrt} [K]$ compared to a no-trees condition? Where are the trees placed (maybe indicate with opaque green circles)?
2. Have you considered other tree positioning approaches, such as projecting error gradients in the U-Net's $T_{mrt}$ prediction on the positions of the trees? That is, somewhat similarly to [[1]](https://arxiv.org/abs/1904.09019), where optimal node-positioning was optimized via gradients in a GNN.
3. What type of trees have been investigated? To what kind of tree does $t_g$ used in your experiments relate?
4. Why does your algorithm suggest to populate east-to-west passages more densely? Wouldn't this prevent wind to transport hot air out of the city in summer? And following up on this, what tree positions does your algorithm predict when exclusively optimizing for winter or summer months?

---

> ### Author Response · Authors · 2023-11-19
> **Re: Official Review of Submission3376 by Reviewer TK26 [1/2]**
>
> We thank the reviewer for taking the time to review our work. We are glad that the reviewer acknowledges that our manuscript is “sound” and that the “results of the experiments support the claims”.
>
> ### Originality; new insights
>
> We would like to kindly remind the reviewer that this work was submitted to an applications track, where the primary goal is to showcase the application of recent deep learning methods to *innovative problem setups*. Unlike the majority of previous works which focus on computational efficiency gains or refining forward modeling, our paper targets a distinctive direction. Our emphasis lies in the use of deep learning to actively optimize an important physical parameter of the real-world, i.e., in our example the improvement of outdoor human thermal comfort. Note that our problem formulation extends beyond this example, i.e., essentially it boils down to making physically meaningful interventions to achieve some goal.
>
> Besides the aforementioned, we would briefly like to note that our work is the first to optimize tree locations to reduce the mean radiant temperature over time spans of weeks or even entire decades. This also allows us to investigate the effect of optimized tree locations over longer time periods (Section 4.3). Overall, we believe that our work may provide an interesting example of how to utilize advancements in deep learning to *actively* support in solving challenging real-world optimization problems.
>
> ### Focus on one city
>
> We openly acknowledge this as a limitation (also refer to our limitations section). However, note that SOLWEIG has been successfully applied to various cities across the world with distinct local climatic regions. Consequently, we are confident that our approach will also work for other climatic regions. However, we hope that the reviewer understands that getting the respective data is unfortunately challenging. Future advancements in estimating the needed data from, e.g., satellite imagery, may overcome this data bottleneck. Lastly, we chose five areas with distinct characteristics and found that our approach works across all of them.
>
> ### W1: Where does the data come from?
>
> We obtained the pre-processed data from Briegel et al [1]; refer for details to them. They were provided with spatial data and obtained past meteorological measurements from the internet. The input data is real data (from the point in time it was recorded) but mean radiant temperatures were simulated with SOLWEIG, as recording real measurements of mean radiant temperature is challenging (see below for a brief discussion on why).
>
> Regarding ERA5 data: ERA5 Land data has a spatial resolution of ca. 9km x 9km, nor does it contain mean radiant temperatures. Please refer to [2] for further details. Note that all study areas are within the same ERA5 Land grid cell. This means, all study areas have the same global meteorological forcing data. Thus, we consider it as constant within our 500m x 500m areas.
>
> ### W2: Comparisons to traditional physical models
>
> We compare our modeling with SOLWEIG in Section 4.1. We find that the U-Net to SOLWEIG error (i.e., 1.93K) is smaller  than the previously found errors of SOLWEIG to real sparse measurements (i.e., 4.4K; refer to Table 5 in [1]). Consequently, it indicates that U-Net is a well-suited computational shortcut and does not (or perhaps a negligible) trade-off between modeling performance and efficiency. Note that this is not typically the case when replacing a physical model with some data-driven approach.
>
> ### W3: Questions about SOLWEIG and evaluation
>
> Yes, SOLWEIG is a popular physical simulation model for estimation of mean radiant temperature. We provide details on SOLWEIG in Appendix C.
>
> Regarding the L1 error: it is computed between the U-Net’s and SOLWEIG’s estimation. Note that obtaining real measurements of mean radiant temperatures is difficult and monetarily expensive, as both short- and longwave radiation from six directions need to be measured. In addition, such measurements can only be made for single points, which in turn makes comprehensive evaluation over an entire spatial area challenging.
>
> ### W4: Comparison of #trees vs. positioning (optimization)
>
> We added a comparison of the effect of planting more trees vs. optimizing or not optimizing their position in Appendix F.3. The results highlight the importance of our optimization, i.e., not optimizing would only harness ca. 50% of their potential in reducing mean radiant temperatures.
>
> ---
>
> [1] Briegel, Ferdinand, et al. "Modelling long-term thermal comfort conditions in urban environments using a deep convolutional encoder-decoder as a computational shortcut." Urban Climate 2023.
>
> [2] https://confluence.ecmwf.int/pages/viewpage.action?pageId=74764925

---

> ### Author Response · Authors · 2023-11-19
> **Re: Official Review of Submission3376 by Reviewer TK26 [2/2]**
>
> ### Q1: Figure 1 legend
>
> It is the difference between the status quo to the status after planting an additional 50 trees. We added green crosses in the figures to indicate the tree positions.
>
> ### Q2: Gradient-based optimization
>
> Yes, but it yielded very poor results. We mainly attribute this to the fact the discretization step was causing issues. We believe modeling mean radiant temperature with a GNN may be fruitful for future work and would allow for such an optimization.
>
> ### Q3: Type of tree
>
> We note that current physical simulation models, such as SOLWEIG, are not capable of distinguishing types of tree. We added a list of limitations of the physical model SOLWEIG in Appendix C to make ML experts aware of such limitations that our approach consequently inherits. Our approach would benefit from advances in incorporating such in physical modeling with SOLWEIG.
>
> ### Q4: Discussion of experimental results
> Our study focuses on mean radiant temperature and, thus, does not consider other effects, such as wind, air temperature or humidity. Thus, we outlined this limitation in Section 5 and proposed it as a next step to integrate them. For example, one could extend our approach to UTCI, a popular thermal comfort index that also includes such factors.
>
> We optimized tree positions for winter and summer months and provided the additional results in Appendix F.2. We find that trees are predominantly positioned in larger, open areas and clustered together during the summer, whereas they are more broadly spread over the neighborhood during the winter.

---

### Official Review · Reviewer_eged · 2023-10-31

**Soundness:** 2 fair
**Presentation:** 4 excellent
**Contribution:** 2 fair
**Rating:** 6
**Confidence:** 4

**Summary:**

This paper addresses the problem of optimizing the placement of trees in an urban environment such as to maximize a proxy metric of human thermal comfort. Given that this quantity is expensive to evaluate using e.g. physical simulations, the authors propose training a convolutional neural network to estimate the quantity instead. Furthermore, the estimation is performed in an aggregated fashion, which can yield substantial speed-ups while sacrificing an acceptable amount of accuracy.

The authors pair this estimation of the objective with a heuristic local search technique comprising several components. The method is applied to the optimization of tree placements within a city, and shows improved performance over either component in isolation. Furthermore, there are interesting insights regarding the impact of tree cover on temperature as a function of time, the diminishing impact of additional trees, and the "what-if" scenario concerning the impact of an alternative, optimized placement over the one that currently exists.

**Strengths:**

**S1**. The work addresses a relevant, practical problem of potential societal impact. I found the analyses in 4.3 and 4.4 particularly interesting and insightful.

**S2**. The paper is very well-written, clear, and easy to follow.

**Weaknesses:**

**W1**. In my opinion, the methodological contributions of the paper are thin, and essentially boil down to 1) estimating the aggregated radiant temperatures directly instead of individually and 2) integrating this estimation in a standard local search procedure. These are both fairly straightforward. The "theoretical analysis" in Section 3.1 is extremely tenuous and, in my opinion, should not be branded as such.

**W2**. The evaluation does not include error bars and confidence intervals, which are a must for drawing reliable conclusions from the results, given the stochasticity of the methods (bar greedy search which is deterministic; but given the estimation itself is learned, one could consider an experimental design where the "seed" for the estimation varies in a paired fashion).

**W3**. There is a potential fundamental limitation in the considered objective, which does not include any measure of population density, traffic, or footfall for the areas.

- In my opinion, this is potentially problematic as the optimization procedure may find solutions where trees are placed in places that yield good deltas in the temperature metric, and yet are comparatively less (or not at all) populated.
- Hence, the placements may trivially not lead to decreases in thermal comfort experienced by individuals, given that an area is not actually frequented by people.
- This is potentially alluded to in the text (bottom of page 8) and acknowledged in Section 5, but I think it deserves to be addressed in more depth.
- For example, it may be that the density is sufficiently uniform in the considered grid that this is not an issue. Evidence that the approach does not "game" the objective would also counteract this point.

**Questions:**

**C1**. Another means of strengthening the evaluation is to include other, more competitive baselines from the literature. Is there no such method? Given the improved scalability attained by your method, what is the point at which prior methods (e.g. that estimate the temperature in a more granular fashion) are too slow? Some more extensive benchmarking than the figures quoted in the text in 4.1 would help.

---

> ### Author Response · Authors · 2023-11-19
> **Re: Official Review of Submission3376 by Reviewer eged**
>
> We thank the reviewer for the thorough review and raising constructive criticisms. We are happy that the reviewer finds that our work “addresses a relevant, practical problem of potential societal impact” and some of our analyses “insightful”. Below, we address the concerns and questions of the reviewer.
>
> ### W1: Methodological contribution
>
> Our work aims to formulate a problem that can be solved by deep learning techniques. Compared to previous works in physical modeling with deep learning that (*passively*) typically just aimed at computational efficiency or improved modeling in the forward direction, we instead take an *active* position by optimizing the input s.t. it optimizes a physical measure to achieve a certain goal (backward direction).
>
> ### W2: Error bars
>
> As suggested by the reviewer, we have added error bars for the optimization of trees in our comparison of optimization strategies (Table 1). It clearly shows the superiority of our optimization method above common baselines for black-box optimization.
>
> ### W3: Missing factors, such as population density, etc.
>
> We also agree this is a limitation (as discussed in the limitations section). However, this is not a limitation of our method. Indeed, we can integrate such factors into the objective function if such data (e.g., the population density) is available. Consequently, it can be ensured that trees are planted in areas, in which they effectively improve the thermal comfort of individuals where they reside.
>
> ### C1: Comparisons to prior works; contextualization of the speed-ups
>
> Note that prior work only used areas of sizes of ca. 50m x 80m [1] or 30m x 80m [2], i.e., our study areas are roughly two orders of magnitude larger. Further, they optimized for a maximum number of 8 [1] or 14 [2] trees, whereas we optimized for up to 200 trees. This makes our approach hardly comparable to prior work.
>
> Regarding the computational speed-ups of our method: we further report speed-ups of factors of 30, 17, 44000, and 400000 for the time periods: day, week, year, or decade, respectively, when comparing using our model that directly outputs the aggregated mean radiant temperature to first predicting the mean radiant temperature for each time step and then aggregating over all time steps. Note that using deep learning methods compared to physical modeling with SOLWEIG achieves another factor of ca. 22x [1]. Note that prior works exclusively used physical simulation models. To contextualize the amount of speed-up we gain, let us consider an example: consider that the optimization takes 1 minute with our method and optimizes the tree positions for the time period decade. Consequently, the estimated runtime for SOLWEIG would be 1 min\*400000\*22=ca. 16.73 years.
>
> ---
>
> [1] Wallenberg, Nils, Fredrik Lindberg, and David Rayner. "Locating trees to mitigate outdoor radiant load of humans in urban areas using a metaheuristic hill-climbing algorithm–introducing TreePlanter v1. 0." Geoscientific Model Development 2022.
>
> [2] Stojakovic, Vesna, et al. "The influence of changing location of trees in urban green spaces on insolation mitigation." Urban Forestry & Urban Greening 2020.
>
> [3] Briegel, Ferdinand, et al. "Modelling long-term thermal comfort conditions in urban environments using a deep convolutional encoder-decoder as a computational shortcut." Urban Climate 2023.

---

> > ### Comment · Reviewer_eged · 2023-11-21
> > **Post-rebuttal response to authors**
> >
> > Many thanks for engaging with my points! I think the response to W2 addresses my main concern in terms of correctness, and C1 contextualizes the scalability of the method well. I'd encourage the authors to add further discussion re. C1 to the main text of the paper and spell this out explicitly. W1 and W3 are not easily addressable and remain limitations of the work. I am raising my score to a 6 as a result of the discussion.

---

> > > ### Author Response · Authors · 2023-11-23
> > > **Thank you!**
> > >
> > > Thank you for the follow-up and updated score! We appreciated the discussion and your feedback.

---

### Official Review · Reviewer_bCfL · 2023-11-01

**Soundness:** 3 good
**Presentation:** 4 excellent
**Contribution:** 2 fair
**Rating:** 8
**Confidence:** 3

**Summary:**

The paper proposes an optimization algorithm for tree placement in urban environments to reduce the impact rising temperatures. The optimization algorithm uses a combination of greedy heuristics, genetic algorithm and hill climbing to find optimized placement locations for trees. The optimization relies on accurate thermal modeling of the urban environment, which is typically a physics based simulator. The paper proposes an ML approach to approximate the simulator and speed up optimization.

**Strengths:**

- The paper is well presented, problem statement is clearly stated, and the results are easy to follow
- Use of real-world data
- Claims are backed by theoretical analysis and detailed experiments.

**Weaknesses:**

- Lack of technical description of the algorithm. It is not clear how the hill-climbing algorithm works. Methods use domain-specific terminology that have not been adequately explained for an ML conference reader.
- Application of standard ML methods, novelty is in a narrow application area
- 500m x 500m is not a "large neighborhood". Prior state-of-the-art is not stated.
- No details on the size of the training data, separation of train-test split details is provided.
- It is unclear how the hyper-parameters are tuned. Is there a separate held-out dataset?
- It would be good to see a cost comparison of planting a new tree vs transferring existing tree.

**Questions:**

- What is the correlation value between mean radiant temperature and mortality?
- What is area covered by prior state-of-the-art in tree placement?
- I did not understand how the aggregated performance is worse than point prediction, the absolute number is lower
- Isn't it a good thing that the temperature increases during nights and winters when the environment is colder?
- How can you plant a big tree wherever you want in the city? Won't they need to be small and grow over time? The size of the trees considered is unrealistic.
- How do you ensure the distribution of training data is sufficient? The optimization algorithm can potentially shift the distribution away from the training set.
- Why is a reduction of 0.83K substantial? Please provide sufficient domain context for an average ML scientist can follow.
- Did not understand how you got 20% reduction for 60C T_mrt. No details have been provided.

---

> ### Author Response · Authors · 2023-11-19
> **Re: Official Review of Submission3376 by Reviewer bCfL [1/2]**
>
> We thank the reviewer for taking the time and raising constructive criticisms. We are glad that the reviewer appreciates our presentation and acknowledges that our “claims are backed by theoretical analysis and detailed experiments.” Below, we address the reviewer’s concerns and questions.
>
> ### Additional technical description
>
> We revised the manuscript and, e.g., added missing explanations of the hill-climbing algorithm. We would provide further feedback if the reviewer sees other parts that may require improvement.
>
> ### Novelty in narrow application area
>
> We would like to note that the majority of previous work on physical modeling focused on improving computational efficiency or improved modeling in the forward modeling. Our work instead focuses on finding interventions in the input, i.e., locations of trees, to optimize a physical quantity (backward direction). Our case study extends beyond the scope of the presented application area and may provide an interesting direction for future work in using deep learning approaches to support humans in challenging optimization problems.
>
> ### 500m x 500m is not a large neighborhood; what area is covered by previous work, and missing state-of-the-art
>
> We kindly disagree. Note that prior work only used areas of sizes of ca. 50m x 80m [1] or 30m x 80m [2], i.e., our study areas are roughly two orders of magnitude larger. Further, they optimized for a maximum number of 8 [1] or 14 [2] trees, whereas we optimized for up to 200 trees. This makes our approach hardly comparable to prior work.
>
> ### Missing dataset details and HP tuning
>
> Thanks for pointing this out! We added the missing dataset details in Appendix B. Note that we did not perform hyperparameter tuning. We believe future work could already improve performance by optimizing the hyperparameters.
>
> ### Cost comparison of planting new trees vs. transferring existing trees
>
> We hope the reviewer understands that a cost comparison is out-of-scope for this work and may depend on many parts that are hard to control for. Nonetheless, we would like to point the reviewer to Section 4.4 that showcases that we could significantly better utilize the potential of urban trees by moving them to “better-suited” locations in terms of reduction in mean radiant temperature. This may be of interest in the consideration whether planting new trees or transferring existing ones is the most cost-efficient solution for a given city.
>
> ### Correlation value between mean radiant temperature and mortality
>
> We kindly refer the reviewer to Table 1 and Figure 3 of Thorsson et al. [3] for details.
>
>
> ### I did not understand how the aggregated performance is worse than point prediction, the absolute number is lower
>
> We compared the direct prediction of aggregated mean radiant temperature against making predictions for every time step and then aggregating the results. We did not compare with SOLWEIG’s results, as it is prohibitively expensive for our larger time periods (year & decade), and wanted to stay consistent within our comparisons. Below, we provide comparisons of the results of aggregated performance compared to SOLWEIG for both smaller time periods (day & week) that make clear why making point prediction and then aggregating is slightly better, though computationally significantly more expensive.
>
> | type | day | week |
> | :- | :-: | -: |
> | nonaggregated         | 1.32        | 0.98         |
> | aggregated         | 1.48         | 1.07         |
>
> ### Isn't it a good thing that the temperature increases during nights and winters when the environment is colder?
>
> Yes, this is indeed the case and it is beneficial. The effect can be seen in Figure 3. We did not delve into this deeper, as we chose to focus on mean radiant temperature reduction for combating heat stress.
>
> ---
>
> [1] Wallenberg, Nils, Fredrik Lindberg, and David Rayner. "Locating trees to mitigate outdoor radiant load of humans in urban areas using a metaheuristic hill-climbing algorithm–introducing TreePlanter v1. 0." Geoscientific Model Development 2022.
>
> [2] Stojakovic, Vesna, et al. "The influence of changing location of trees in urban green spaces on insolation mitigation." Urban Forestry & Urban Greening 2020.
>
> [3] Thorsson, Sofia, et al. "Mean radiant temperature–A predictor of heat related mortality." Urban Climate 2014.

---

> > ### Author Response · Authors · 2023-11-19
> > **Re: Official Review of Submission3376 by Reviewer bCfL [2/2]**
> >
> > ### How can you plant a big tree wherever you want in the city? Won't they need to be small and grow over time? The size of the trees considered is unrealistic.
> >
> > This is a simplifying assumption. As noted in our limitations sections, there are many other factors one has to consider that make the optimization problem substantially more challenging in practice. We note that we assumed a static urban environment (footnote 4), including static vegetation, i.e., trees are assumed to not grow over time. This is a common assumption due to the difficulty of modeling such dynamic processes. Note that the physical model SOLWEIG also makes this assumption and, as a consequence, our approach also inherits this limitation. We make this limitation stemming from the physical modeling more clear and added a list of limitations for SOLWEIG in Appendix C to make ML experts more aware of the limitations of physical modeling and, consequently, also our method. Finally, note that the tree size is a changeable parameter and depends on the user. We note that such trees are realistic in the considered study area; even though we admit they belong to the top 20-percentile.
> >
> > ### How do you ensure the distribution of training data is sufficient?
> >
> > This is a great point! We trained the model on sufficient data (Appendix B). Further, we added an analysis in Appendix F.1 to confirm that the optimized tree locations actually lead to similar effects in the physical modeling with SOLWEIG and are not just mere artifacts of out-of-distribution changes. It confirms that it is the former, highlighting that the found solution is not a mere artifact of the deep learning method.
> >
> > ### Why is a reduction of 0.83K substantial?; Did not understand how you got 20% reduction for 60C T_mrt.
> >
> > The reduction is indeed substantial. We recognize that understanding the figure for non-domain experts may be challenging. To enhance the understanding, we also analyzed the change in peak instances, specifically the reduction in hours with heat stress, i.e., mean radiant temperature of above 60°C. To this end, we summed up all such hours of the entire study areas and calculated the reduction before and after the optimization. Our counterfactually-inspired optimization in Section 4.4 yielded a remarkable 20% decrease in hours during the hottest week in 2020. Note that  the number of trees stays the same in this experiment and no new trees are added.
> >
> > To provide a deeper perspective, consider that only 6–8 hours a day (from late morning to late afternoon) typically have mean radiant temperature exceeding 60°C. During all other hours, including nighttime, mean radiant temperatures usually remain below this threshold. Consequently, the 20% reduction is confined to 8 hours a day, or a total of 56 hours a week. In practical terms, the 20% reduction translates to ca. 11.2 hours per week, or an average of 1.6 hours per day of reduced hours exceeding the threshold. This means an additional average of 90 minutes each day when individuals can enjoy a more comfortable outdoor experience in terms of thermal comfort.

---

> > > ### Comment · Reviewer_bCfL · 2023-11-22
> > > **Thank you for responses**
> > >
> > > The responses have satisfied the questions I had raised and significantly addresses the weaknesses. I'll raise my score.
> > >
> > > I do agree with one of the other reviewers that this paper is not suited for ICLR even if it is accepted, an ML audience will not appreciate the contributions here. It is better suited for urban planning venues.

---

> > > > ### Author Response · Authors · 2023-11-23
> > > > **Thank you!**
> > > >
> > > > Thank you for the follow-up and updated score! We appreciated the discussion and your suggestions.
> > > >
> > > >
> > > > Concerning the venue, we believe that ICLR is a good platform for our work. Our aim is to prompt ML experts leveraging their skills across various scientific domains to reconsider the prevailing emphasis mostly exclusively on computational efficiency or general forward modeling. Instead, we would be happy if the takeaway of our paper for them would be to consider alternative ML applications in the scientific domain more often, such as optimization of some physical quantity.

---

### Meta-Review · Area_Chair_gsMv · 2023-12-10

**Metareview:**

This paper introduces a set of algorithms for the mitigation of urban heat island effects from climate change through optimal placement of trees. The performance of the method is shown to exceed an existing physics-based baseline. The reviewers are divided on whether this paper is suitable for ICLR. This is tricky, since it's very hard for applied papers to find a home at ICLR, and as the authors correctly state it is vitally important for such papers to become more mainstream in this venue. In this case, however, I must recommend rejection, since here there are three factors that together reduce the applicability to ICLR: 1) a problem that isn't yet well-established within the ML community, 2) fairly standard methods from an ML perspective, and 3) lack of other strong ML baselines for comparison. One, or perhaps two, of these points could hold true and I would fight for the paper to be accepted at ICLR. However, with the combination of all three, I think that it is more appropriate for a different venue. I would suggest the authors consider for example the top journal Environmental Data Science as a way to reach the research community at the intersection of machine learning and climate action.

**Justification For Why Not Higher Score:**

See the meta-review.

**Justification For Why Not Lower Score:**

n/a

---

### Decision · Program_Chairs · 2024-01-16

Reject